# Environmental sensing by mature B cells is controlled by the transcription factors PU.1 and SpiB

Simon N. Willis[1,2], Julie Tellier[1,2], Yang Liao[1,2], Stephanie Trezise[1,2], Amanda Light[1], Kristy O'Donnell[1,3], Lee Ann Garrett-Sinha[4], Wei Shi[1,5], David M. Tarlinton[1,2,3] & Stephen L. Nutt [1,2]

Humoral immunity requires B cells to respond to multiple stimuli, including antigen, membrane and soluble ligands, and microbial products. Ets family transcription factors regulate many aspects of haematopoiesis, although their functions in humoral immunity are difficult to decipher as a result of redundancy between the family members. Here we show that mice lacking both PU.1 and SpiB in mature B cells do not generate germinal centers and high-affinity antibody after protein immunization. PU.1 and SpiB double-deficient B cells have a survival defect after engagement of CD40 or Toll-like receptors (TLR), despite paradoxically enhanced plasma cell differentiation. PU.1 and SpiB regulate the expression of many components of the B cell receptor signaling pathway and the receptors for CD40L, BAFF and TLR ligands. Thus, PU.1 and SpiB enable B cells to appropriately respond to environmental cues.

[1] The Walter and Eliza Hall Institute of Medical Research, 1G Royal Parade, Parkville, VIC 3052, Australia. [2] The Department of Medical Biology, University of Melbourne, Parkville, VIC 3010, Australia. [3] Department of Immunology and Pathology, Monash University Level 6, Burnet Tower 89 Commercial Road, Melbourne, 3004 VIC, Australia. [4] Department of Biochemistry, State University of New York at Buffalo, Buffalo, NY 14203, USA. [5] Department of Computing and Information Systems, University of Melbourne, Parkville, VIC 3010, Australia. Correspondence and requests for materials should be addressed to S.N.W. (email: willis@wehi.edu.au) or to S.L.N. (email: nutt@wehi.edu.au)

Antibody-mediated immunity relies on the ability of B cells to respond to multiple environmental stimuli including antigen, Toll-like receptor (TLR) ligands, and T-cell-derived help, including CD40L and the cytokines interleukin-4 (IL-4) and IL-21. The survival of mature B cells and plasma cells also depends on members of the tumor necrosis factor receptor superfamily (TNFRSF), including the B-cell-activating factor receptor (BAFF-R)[1]. Mature B cells, including follicular and marginal zone (MZ) B cells, are quiescent and relatively long lived. After exposure to cognate antigen, B cells re-enter the cell cycle and undergo multiple rounds of division, as well as initiating immunoglobulin class switch recombination (CSR)[2]. Proliferating B cells have the potential to differentiate into short-lived plasmablasts that provide the immediate, but low affinity, antibody that is important early in the immune response. Alternatively, in response to antigen and T cell help, activated B cells can enter a structure termed the germinal center (GC), where they undergo clonal amplification and somatic hypermutation and differentiation into plasma cells that secrete high-affinity antibodies[2]. GCs also produce memory B cells that can rapidly differentiate into plasma cells upon re-exposure to antigen.

A complex network of transcription factors controls each aspect of the B cell response to antigen. This network includes factors that are essential for B cell proliferation and the GC response, including PAX5, BACH2, IRF4/BATF, IRF8, NFκB, E-proteins (E2A, E2-2) and Oct2/OBF1, whereas a smaller group, including high concentrations of IRF4, BLIMP-1/PRDM1, ZBTB20 and XBP1, are required for plasma cell differentiation and antibody production (reviewed in refs. [3–5]). We have reported a role for a complex of the transcription factors PU.1 and IRF8 in negatively regulating plasma cell differentiation in cell culture, although the role of these factors in vivo is unclear[6, 7].

The Ets family transcription factor PU.1, encoded by the *Spi1* gene, is a major regulator of haematopoiesis, controlling the expression of hundreds of genes including growth factor receptors, adhesion molecules, transcription factors and signaling components[8]. PU.1-deficient mice lack all lymphocytes, including B cells, suggesting that PU.1 is an essential regulator of the B cell developmental pathway[9–12]; however, this requirement is limited to early lymphopoiesis as conditional deletion of PU.1 in CD19-expressing B cells is compatible with normal development and function[10, 13–16]. This minimal consequence of PU.1 loss in B cells is surprising, as PU.1 is well-known to bind tens of thousands of sites in the B cell genome. One possible explanation for this discrepancy is the strong expression of SpiB, the most closely related Ets family member in B cells, that binds to the identical nucleotide sequence GGAA[17, 18]. Indeed, *Spi1*[+/−]*Spib*[−/−] mice have markedly reduced B-cell numbers[19], and inactivation of PU.1 and SpiB in bone marrow B-cell progenitors results in a block in development at the pre-B-cell stage and the formation of pre-B acute lymphoblastic leukemia (ALL)[15]. A similar developmental block and leukemogenesis is also observed in B cells lacking PU.1/IRF4 or IRF4/IRF8, demonstrating that an Ets/IRF complex controls bone marrow B-lymphopoiesis[13, 20].

To better define the function of PU.1 and SpiB in mature B cells, we here utilize the *Cd23*-Cre strain[21] to inactivate PU.1 in *Spib*[−/−] mice at the transitional B-cell stage in the spleen, thus allowing us to bypass the developmental block that occurs when both PU.1 and SpiB are deleted in immature B cells. We find that PU.1/SpiB redundantly control mature B cell survival and differentiation in response to a diverse range of stimuli. PU.1/SpiB-deficient B cells cannot signal efficiently through the B-cell receptor (BCR) and thus mutant mice lack antigen-specific B-cell responses entirely. PU.1 and SpiB directly control many components of the BCR pathway as well as important receptors for T-cell-derived signals and microbial products. In keeping with this observation, PU.1/SpiB-deficient B cells are not very responsive to these stimuli in vitro, but plasma cell lineage differentiation is increased, a cellular stage in which *Spi1* is lowly expressed and the *Spib* gene is silenced. These findings highlight PU.1 and SpiB as cell intrinsic regulators of B cell responsiveness to environmental cues, a critical process for humoral immunity.

## Results

**PU.1 and SpiB control follicular B cell homeostasis.** To investigate the function of PU.1 and SpiB in mature B cells we have generated mice that carry floxed alleles of *Spi1*[14], null alleles of *Spib*[22] and a BAC transgene where Cre recombinase is brought under the control of the *Cd23* (*Fcer2*) regulatory sequences (*Spi1*[fl/fl]*Spib*[−/−]*Cd23*-Cre[T/+]), hereafter termed PU.1 SpiB double knockout (DKO). *Cd23*-Cre is active from the transitional stage of B-lymphopoiesis, which results in highly efficient gene inactivation in both follicular and MZ B cells[21]. The frequency and phenotype of splenic B cells from the PU.1 SpiB DKO mice was compared to control (*Cd23*-Cre), PU.1 conditional knockout (cKO; *Spi1*[fl/fl]*Cd23*-Cre[T/+]) and SpiB knockout (KO; *Spi1*[fl/fl]*Spib*[−/−]*Cd23*-Cre[+/+]) genotypes. Although the singular loss of either PU.1 or SpiB did not substantially alter the proportion, total number or subset make up of the splenic B cell compartment, the proportion and number of follicular B cells was decreased 2-fold to 3-fold in PU.1 SpiB DKO spleens (Fig. 1a, b), whereas the number of MZ B cells remained unaltered. In keeping with a previous report[23], PU.1 and SpiB were redundantly required for cell surface expression of CD23, which was significantly down-regulated in the absence of SpiB (Supplementary Fig. 1) and more so in the absence of both factors (Fig. 1a). As a consequence, follicular B cells were defined as B220[+]IgD[+]IgM[int] (Fig. 1a). Taken together, these data suggest that PU.1 and SpiB are redundantly required for follicular B cell survival.

**Increased plasma cells without PU.1 and SpiB.** In contrast to the relative loss of follicular B cells, the combined deficiency of PU.1 and SpiB resulted in a significantly increased frequency and proportion of plasma cells (B220[−]CD138[+]CD98[+]) in both the spleen and bone marrow (Fig. 2a, b). PU.1 cKO mice displayed a normal plasma cell frequency while SpiB KO mice harbored significantly more plasma cells in the spleen, with a similar but non-significant increase in the bone marrow. Analysis of the serum immunoglobulin titers and isotype make up of the plasma cell compartment revealed a significant increase in IgM and IgG3 in SpiB KO and PU.1 SpiB DKO sera and decreased IgA and IgG2b (Supplementary Fig. 2). Curiously, PU.1 cKO mice displayed increased titers of IgG2b and decreased titers of IgG2c that were not seen in the PU.1 SpiB DKO mice. In keeping with the increase in IgM and IgG3, isotypes that arise from extrafollicular sources, such as MZ B cells[24], immunofluorescence on the spleens of naive PU.1 SpiB DKO mice revealed a marked increase in IgM[+] plasma cells in the red pulp (Fig. 2c). Taken together these data demonstrate that PU.1 and SpiB act both individually and redundantly to negatively regulate plasma cell homeostasis.

**The germinal center reaction requires PU.1 and SpiB.** Previous studies found that SpiB was dispensable for the establishment of a GC, but that the GC could not be sustained beyond 4 weeks[22], while mice lacking a single allele of *Spi1* and both copies of *Spib* throughout B-cell development generated few mature B cells that could not initiate a GC reaction[19]. However, in neither study was the fate of the antigen-specific B cells tracked.

Analysis of control mice 14 days after immunization with the T cell dependent antigen NP-KLH in alum revealed robust production of NP-binding B cells that had undergone CSR to

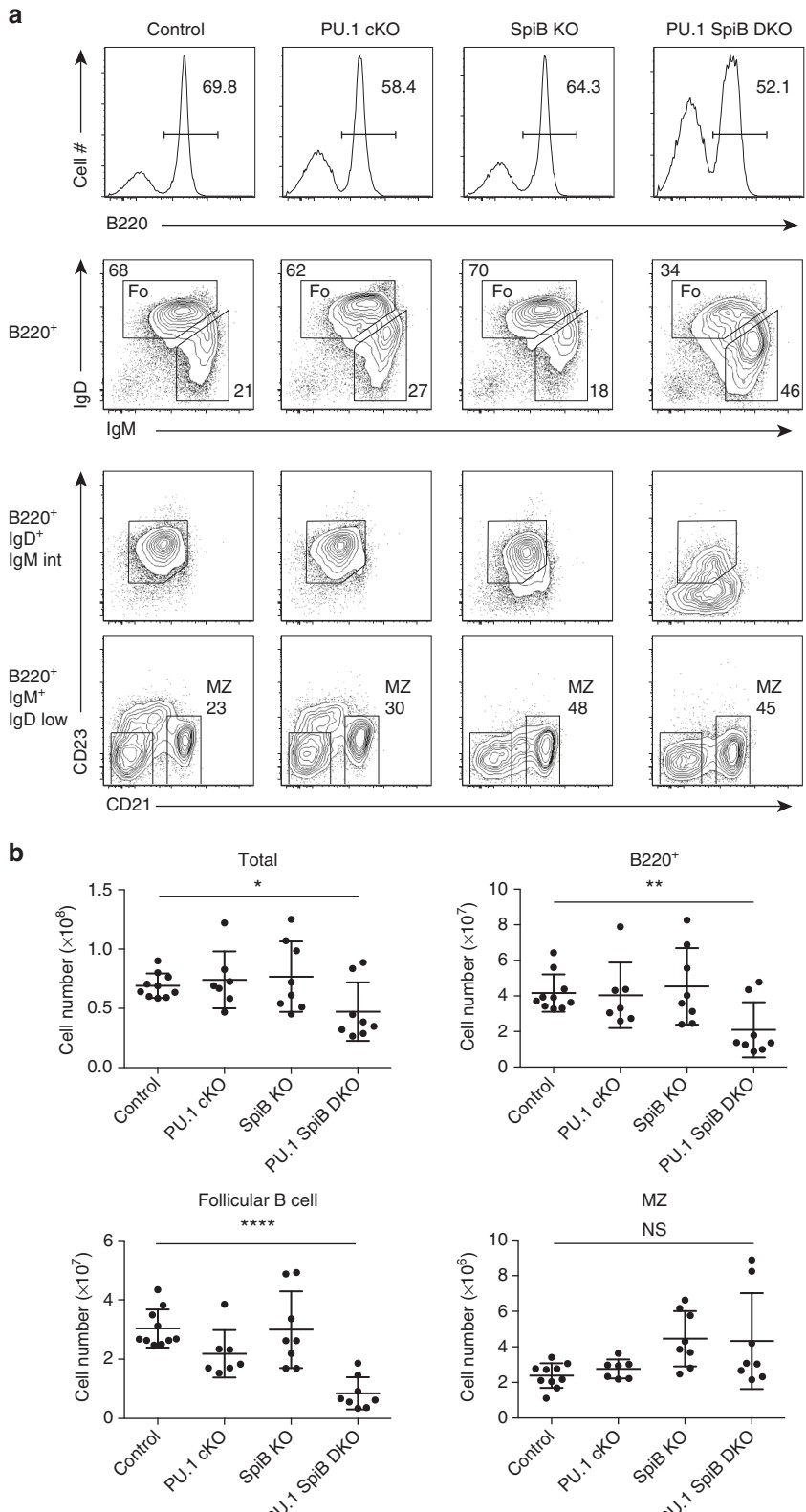

**Fig. 1** PU.1 and SpiB regulate follicular B cell numbers. **a** Flow cytometric analysis of splenocytes from control (*Cd23*^T/+), PU.1 cKO (*Spi1*^fl/fl *Cd23*^T/+), SpiB KO (*Spi1*^fl/fl *Cd23*^+/+ *Spib*^−/−) and PU.1 SpiB DKO (*Spi1*^fl/fl *Cd23*^T/+ *Spib*^−/−) mice showing the abbreviated gating strategy for the detection of follicular (Fo) B cells (B220^+ IgD^+ IgM^int) and MZ (B220^+ IgM^+ IgD^low CD21^+) B cells. As the loss of PU.1 and SpiB leads to the downregulation of CD23 expression, follicular B cell numbers were based on the gate for B220^+ IgD^+ IgM^int. The full gating strategy is shown in Supplementary Fig. 8A. Data are representative of 7–10 mice per genotype. **b** Absolute cell numbers are graphed as the mean ± s.d. *P*-values compare the indicated samples (two tailed *t*-test). Each circle represents the results from an individual mouse. \**P* < 0.05, \*\**P* < 0.01, \*\*\*\**P* < 0.0001

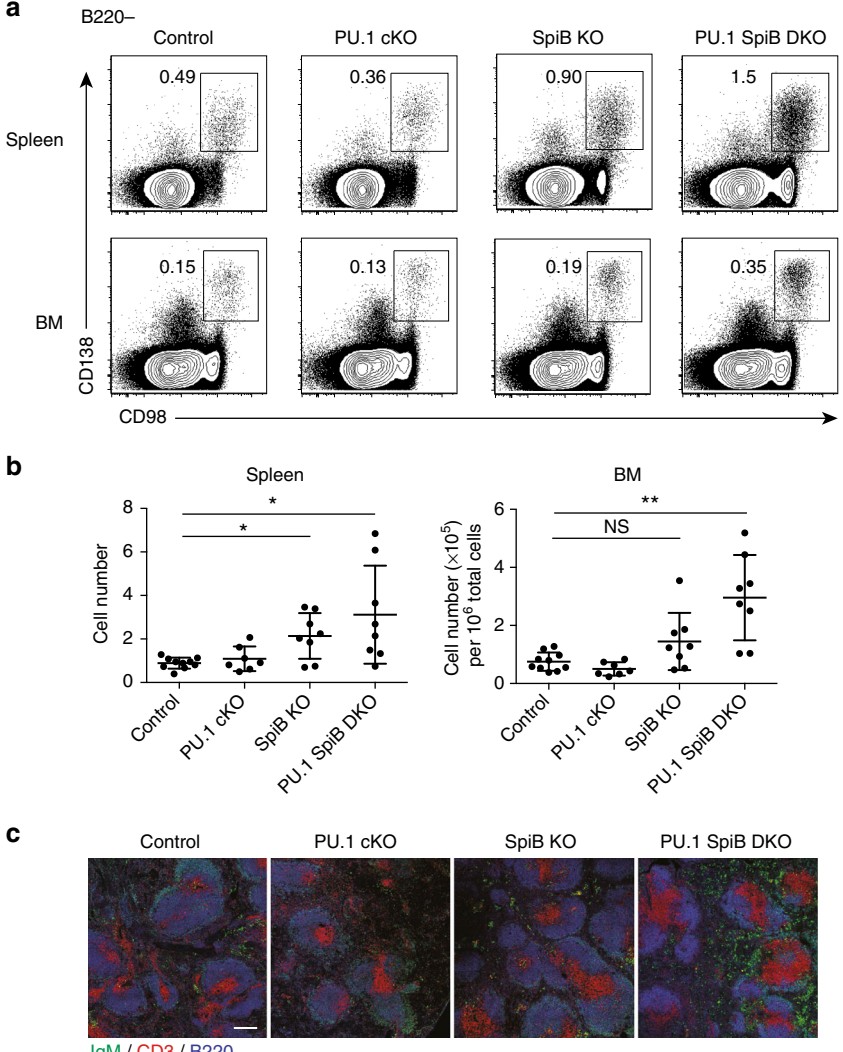

**Fig. 2** Increased plasma cell frequency in PU.1 SpiB DKO mice. **a** Flow cytometric analysis of splenocytes and bone marrow from mice of the indicated genotype for the detection of plasma cells (B220$^-$ CD138$^+$ CD98$^+$). Data are representative of 7–10 mice per genotype. The full gating strategy is shown in Supplementary Fig. 8B. **b** Absolute numbers (for spleen) and proportion (for bone marrow) of plasma cells are graphed as the mean ± s.d. *P*-values compare the indicated samples (two tailed *t*-test). Each circle represents the results from an individual mouse. *$P < 0.05$, **$P < 0.01$. **c** Immunofluorescence of spleen sections of each of the four genotypes stained for T cells (CD3, red), B cells (B220, blue) and antibody-secreting cells (IgM$^{high}$, green). Scale bar = 200 μm

IgG1 and near uniformly upregulated the GC regulator Bcl6. (Fig. 3a, b). As expected SpiB KO B cells responded similarly to controls at this time point. In contrast, immunization of PU.1 SpiB DKO mice elicited virtually no response, generating neither IgG1$^+$ nor Bcl6$^+$ GC B cells (Fig. 3a, b, d). PU.1 cKO, in contrast to our previous studies using *Cd19*-Cre[14], displayed an intermediate phenotype, with reduced NP$^+$IgG1$^+$ and NP$^+$ Bcl6$^+$ GC B cells compared to the control mice. Immunofluorescence studies of spleen sections from immunized mice confirmed the presence of GC structures in control mice and those singly deficient in PU.1 or SpiB, but their absence in PU.1 SpiB DKO mice (Fig. 3e). Similarly, ELISpot analysis of the NP-specific antibody response revealed a normal frequency of NP-binding IgG1 antibody-secreting cells at day 14 post immunization in mice lacking SpiB, a non-significant reduction in mice lacking PU.1 and very few NP reactive cells in the DKO spleen (Fig. 3c, d). Together these data indicate that PU.1 and SpiB are redundantly required for the GC reaction and the production of high-affinity antibody.

**PU.1 and SpiB negatively regulate plasma cell differentiation**. Our previous studies using *Cd19*-Cre to inactivate PU.1 revealed that PU.1 and IRF8 acted together to repress plasma cell differentiation in vitro[6]. It that model, the singular deficiency of PU.1 resulted in a relatively mild enhancement of plasma cell formation after stimulation with CD40 L and IL-4, a finding that agreed with a second study in which inactivation of a miRNA binding site in the 3′ untranslated region of *Spi1* resulted in an increased concentration of PU.1 in activated B cells and impaired plasma cell formation[25].

To address the combined importance of PU.1 and SpiB for B cell differentiation in vitro we cultured follicular B cells of the four genotypes, generated exclusively from lymph nodes to exclude any possible MZ B cell contamination, in CD40 L + IL-4, conditions that promote B cell proliferation, CSR to IgG1 and plasma cell differentiation[26]. Although B cells of each genotype proliferated similarly in CD40 L + IL-4, as assessed by CTV dilution (Fig. 4a), the yield of cells was consistently reduced in the PU.1 cKO and PU.1 SpiB DKO cultures (Fig. 4b). Analysis of the

frequency and total number of CD138+ plasma cells and IgG1+ B cells displayed an inverse relationship in that PU.1 cKO and PU.1 SpiB DKO cultures had increased plasma cells and reduced IgG1+ B cell numbers compared to the control and SpiB KO cultures.

**Identification of PU.1 and SpiB dependent genes in B cells**. To determine how the singular and combined loss of PU.1 and SpiB affected B cell differentiation, we isolated RNA from lymph node follicular B cells of the four genotypes before and after 48 h of culture in the presence of CD40 L + IL-4 and performed RNA-

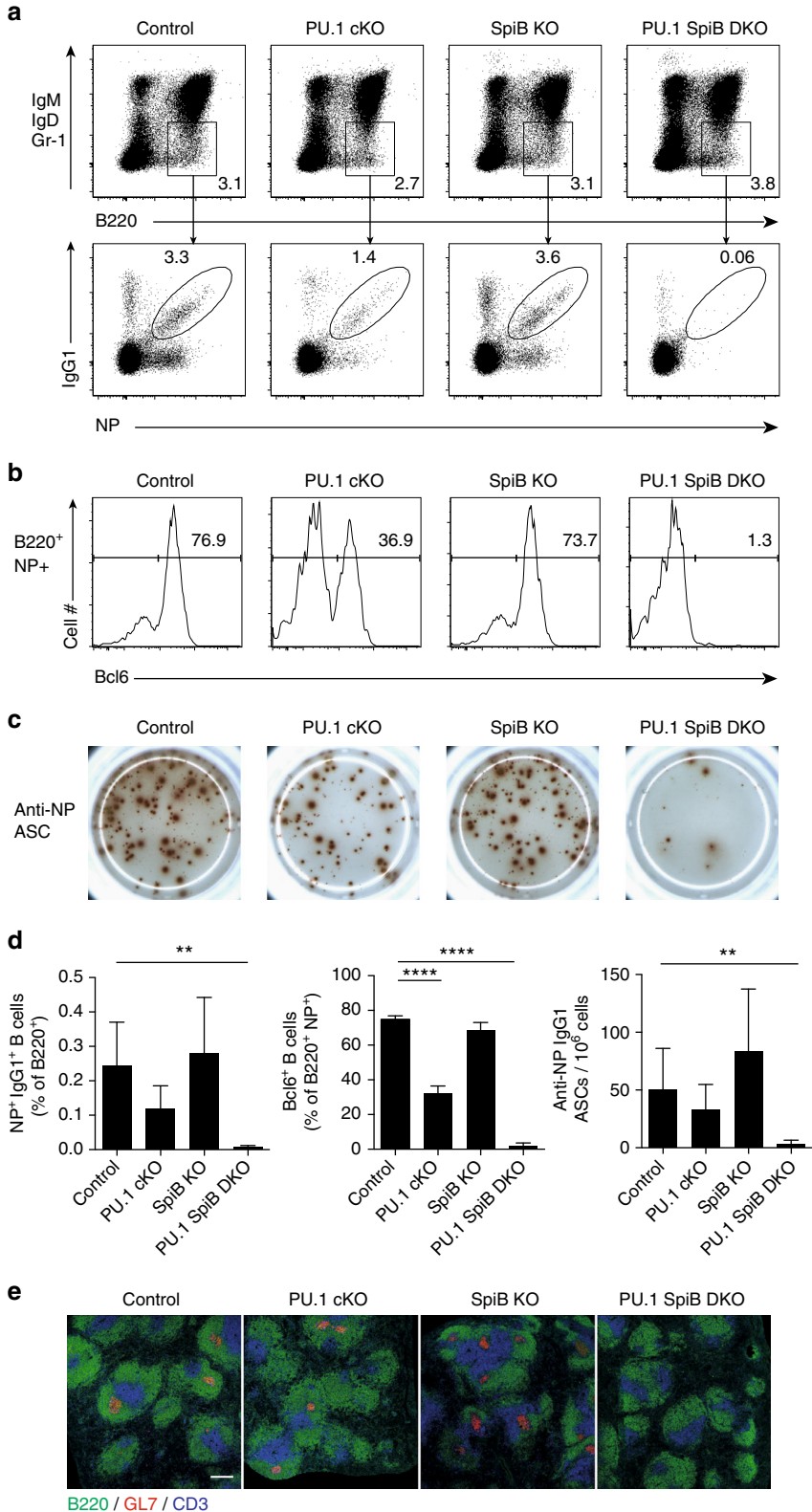

seq. Analysis of the frequency of the inactivation of the floxed *Spi1* exon 5 in follicular B cells displayed complete inactivation in PU.1 cKO B cells, but marked retention of exon 5 in PU.1 SpiB DKO B cells (Supplementary Fig. 3A). This may reflect a reduction in the expression of Cre, which is under the control of the *Cd23* promoter in SpiB-deficient cells (Supplementary Fig. 1) or a requirement by follicular B cells to retain a minimal level of either PU.1 or SpiB for survival. Differential expression (DE) analysis revealed that PU.1 and SpiB regulated 547 and 332 genes, respectively, in naive follicular B cells (Supplementary Fig. 3B), with a clear redundancy observed in the absence of SpiB and a partial deletion of PU.1 (1306 DE genes). Pathway analysis revealed association of genes whose expression required either PU.1 or SpiB with a follicular B-cell signature (Supplementary Fig. 3C, D), while genes repressed by SpiB were uniquely associated with other cell lineages including plasmacytoid dendritic cells (pDC), macrophages and CD8 T cells (Supplementary Fig. 3C). *Tnfrsf13c*, encoding the BAFF-R, which is essential for peripheral B cell survival and homeostasis was significantly downregulated at both the transcript (Supplementary Fig. 4A) and protein level (Supplementary Fig. 4B, C)[1].

Given the partial retention of PU.1 observed in naive follicular B cells, we turned our attention to the RNA-seq data that was generated in activated B cell cultures. At this early time point following activation, we observed minimal differentiation into CD138[+] plasmablasts (<1%; Supplementary Fig. 5A) and a modest increase in proliferation in PU.1 cKO and PU.1 SpiB DKO B cells as judged by CTV labeling (Supplementary Fig. 5B). Importantly, exon 5 of *Spi1* was completely excised regardless of the presence or absence of *Spib* (Fig. 5a), confirming that neither factor was absolutely essential for B cell proliferation and survival following activation in vitro. A similarly high deletion efficiency was obtained in activated B cell cultures stimulated with LPS + IL-4 (Supplementary Fig. 6C), where enhanced B cell differentiation into plasmablasts was also observed (Supplementary Fig. 6A, B).

Analysis of the RNA-seq data derived from CD40 L + IL-4 activated B cells revealed that the loss of PU.1 alone or in combination with SpiB resulted in 1009 and 1443 DE genes, respectively (Fig. 5b and Supplementary Data). In contrast, SpiB deficiency had minimal impact on the transcriptome of activated B cells (17 DE genes, Fig. 5B and Supplementary Data), potentially due to the rapid downregulation of *Spib* expression during B-cell differentiation. Our re-analysis of ChIPseq data of PU.1 binding in LPS activated plasmablasts found that PU.1 bound in 4826 genes[27]. Comparison with the gene expression data revealed a highly significant enrichment of PU.1 binding in genes that required PU.1 or PU.1 and SpiB for their expression (Fig. 5c and Supplementary Data). These data demonstrate that PU.1 functions predominantly as an activator of gene expression during B cell terminal differentiation.

To investigate how PU.1 and SpiB interact with the gene regulatory networks controlling late B cell and plasma cell differentiation, we interrogated our RNA-seq data from activated B cells for the expression of the transcriptional regulators known to be important in these processes[5]. Although all components of the B cell network were expressed in the absence of PU.1 and SpiB, indicating that the network is intact at a global level, we observed partial reduction in several key regulators, including *Pax5* and *Bcl6* in PU.1 cKO and PU.1 SpiB DKO B cells (Fig. 5d). PU.1 and SpiB also appeared to redundantly regulate the expression of *Bach2*, a critical negative regulator of *Blimp1* and plasma cell differentiation[28]. In contrast the expression of other important regulators, including *Irf4*, *Irf8*, and *Pou2af1* (Obf1) was similar in all genotypes. Curiously, we observed a small but significant increase in *Tcf3* (E2A) and a similar decrease in *Tcf4* (E2-2) in PU.1 SpiB DKO B cells (Fig. 5d). Similar results were also observed in activated B cell cultures following stimulation with LPS + IL-4 (Supplementary Fig. 6D).

Analysis of the transcriptional regulators of the plasma cell fate identified premature de-repression of *Prdm1*, *Ell2*, *Irf4* and *Cebpb* without PU.1 or PU.1 and SpiB and *Xbp1* in the absence of both factors (Fig. 5d). As no bona fide plasma cells are observed at this early time point (Supplementary Fig. 3), we interpret the de-repression of these genes to be early markers of the increased propensity of the PU.1 cKO and PU.1 SpiB DKO B cells to undergo terminal differentiation. Notably, we also observed a robust increase in *Aicda*, encoding Activation induced cytidine deaminase (AID) in PU.1 cKO and PU.1 SpiB DKO B cells, making it unlikely that a deficiency in this enzyme is the underlying cause of the defective CSR observed in vitro and in vivo without PU.1 and SpiB (Fig. 5d, Supplementary Fig. 6D).

Gene ontology analysis of the DE genes revealed enrichment for "Metabolic pathways" in each gene set. More informatively the pathways downregulated in PU.1 cKO and PU.1 SpiB DKO largely overlapped and included several gene sets associated with lymphocyte signaling, including the BCR signaling pathway (Fig. 5e). Genes upregulated showed a strong increase in gene frequency in the DKO comparison and included the "protein processing" gene set that contains many genes involved in the unfolded protein response, which is critical for high level antibody secretion by plasma cells (Fig. 5e).

Given the strong deficiency we observed in PU.1 SpiB DKO mice after T-dependent immunization and the existing literature in this area, we decided to focus on the B cell receptor (BCR) signaling pathway. Strikingly, although the core constituents of the BCR (*Igh*, *Igk*, *Igl*, *Cd79a*, and *Cd79b*) were normally expressed, many downstream components of the BCR signal transduction cascade were significantly downregulated (most between 2-fold and 3-fold) in the absence of PU.1 and SpiB (Fig. 6a). A similar trend, albeit with a lower fold change was also observed in PU.1 cKO B cells. Western analysis confirmed these findings, with modest reductions in protein expression for several BCR signaling components including Btk, Rasgrp3, and Card11 (Fig. 6c) in the absence of PU.1 or PU.1 and SpiB. Analysis of the PU.1 ChIPseq data identified the presence of PU.1 binding in the proximal promoter of most of the PU.1 and PU.1/SpiB dependent components of the BCR pathway (Fig. 6b). A similar trend, in

**Fig. 3** Deletion of PU.1 and SpiB abrogates the GC response. Mice of the indicated genotypes were immunized with NP-KLH in alum and analyzed 14 days later. **a**, **b** Flow cytometric analysis of splenocytes. Isotype switched B cells (B220[+]IgM[−] IgD[−]Gr-1[−]) were analyzed for **a** NP[+] IgG1[+] status and **b** Bcl6. Numbers in **a**, **b** are the proportion of cells in the indicated gates. The full gating strategies are shown in Supplementary Fig. 8C, D. **c** Representative ELISPots show the number of NP-specific IgG1[+] antibody-secreting cells (ASC) per million splenocytes plated. Data are representative of 5 or 6 mice per genotype from 2 independent experiments **d** Frequencies of NP[+] IgG1[+] B cells are shown relative to total B220[+] cells (left), proportion of B220[+] NP[+] cells expressing Bcl6 (middle), frequency of total NP-specific IgG1 secreting ASC in the spleen per million cells (right) are shown. Data are graphed as mean ± s. d. of between five and six mice per genotype from two independent experiments. *P*-values compare the indicated samples (two tailed *t*-test). **$P < 0.01$, ****$P < 0.0001$. **e** Spleen sections from mice of each of the genotypes examined mice were stained with antibodies to CD3 to detect T cells (blue), B220 to identify B cell follicles (green) and GL7 for GCs (red). Scale bar = 200 μm

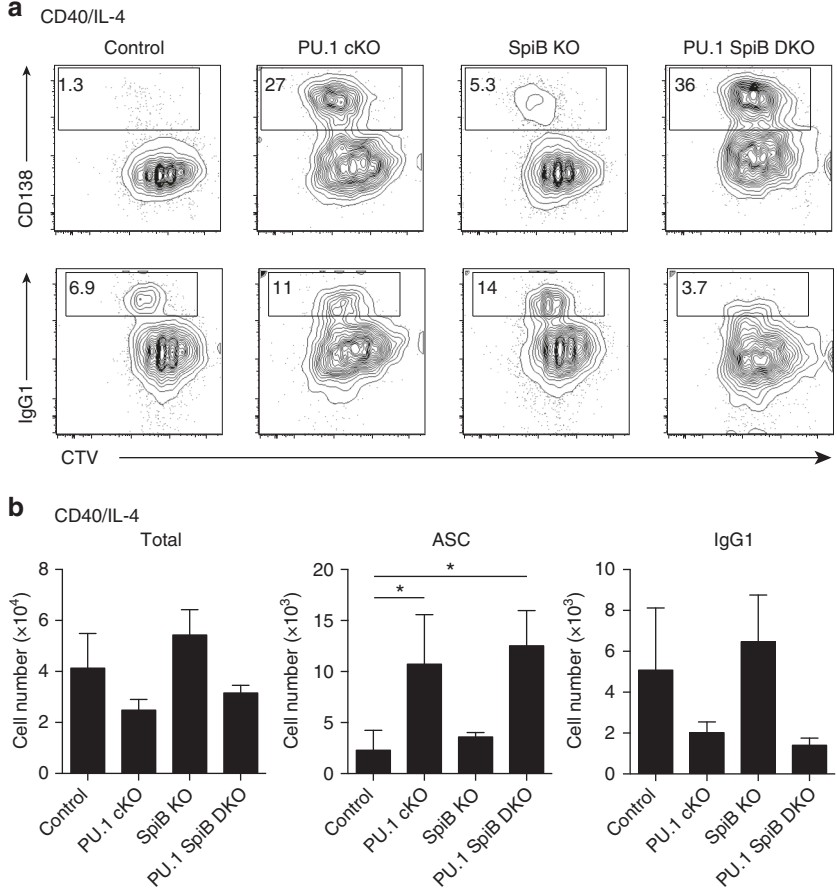

**Fig. 4** PU.1 and SpiB repress antibody-secreting cell differentiation in vitro. **a** Resting lymph node B cells from mice of the indicated genotype were labeled with CTV and cultured in the presence of CD40 L + IL-4 for 4d before flow cytometric analysis. Numbers in the boxes are the proportion of CD138[+] ASC or IgG1[+] B cells. The full gating strategy is shown in Supplementary Fig. 9A. Data are representative of three independent experiments. **b** Quantitation of the total cell number (left), total number of ASCs (middle) and IgG1[+] cells (right) cultured as in **a** is shown after 4d of culture. Data are the mean ± s.d. for three independent experiments. *P*-values compare the indicated samples (two tailed *t*-test). *$P < 0.05$

several BCR signaling components was also observed following activation with LPS + IL-4 (Supplementary Fig. 7).

To determine if these quantitative changes in the expression of multiple components of the BCR resulted in impaired signaling and cellular responses, we cultured follicular B cells of the four genotypes in the presence of anti-IgM + IL-4 for 3 days. Proliferation was similar between the control, PU.1 cKO and SpiB KO B cells, however in keeping with the gene expression data, PU.1 SpiB DKO B cells displayed extremely low viability after BCR engagement (Fig. 6d). To assess more proximal signaling events after BCR crosslinking we could not use the traditional cell source of naive B cells as the inactivation of *Spi1* is incomplete in PU.1 SpiB DKO follicular B cells (Supplementary Fig. 3A). To circumvent this problem, we first cultured naive follicular B cells for 48 h in CD40 L + IL-4, a time point where exon 5 of *Spi1* was absent (Fig. 5a), rested them for one hour and then exposed the cells to anti-IgM. This approach showed clear inhibition in the BCR response, as assessed by calcium mobilization, in the absence of PU.1, and to a lesser extent SpiB (Fig. 6e). These data demonstrate that PU.1 and SpiB are redundant regulators of multiple components of the BCR signaling pathway.

**Regulation of immune receptors by PU.1 and SpiB.** In addition to the regulation of the genes encoding members of the BCR signaling pathway, it was striking that PU.1 or PU.1 and SpiB

were required for the normal expression of several other immune receptors critical for the ability of a B cell to sense its immunological environment (Fig. 7a, Supplementary Fig. 7B). This included *Tnfrsf13c*, encoding the BAFF-R, essential for peripheral B cell survival and homeostasis[1] and *Cd40*, a critical component of T cell help[29]. Flow cytometric analysis confirmed the downregulation of both the BAFF-R and CD40 in the absence or PU.1, with a further reduction in expression in the absence of both PU.1 and SpiB (Fig. 7b, c). Analysis of the PU.1 ChIP-seq data again showed the presence of PU.1 binding in the proximal promoter of both of these targets (Fig. 7d).

This quantitative reduction in *Tnfrsf13c* is likely to be responsible for the 2-fold to 3-fold reduction in follicular B cells in PU.1 SpiB DKO mice (Fig. 2b), while the deceased CD40 expression may be responsible for the lower cell yields in cultures containing CD40 L and IL-4, despite the increased propensity to differentiate into ASCs (Fig. 4b). In contrast, cytokine receptors, such as the *IL4ra* and *Il5r* were marginally upregulated in the absence of PU.1 and SpiB (1.6-fold each).

Several important microbial sensors were also directly regulated by PU.1 and SpiB including *Tlr4* and *Tlr9*, with each having clear proximal promoter-binding sites for PU.1 (Fig. 7a, d, Supplementary Fig. 7B). To examine the relevance of the reduced TLR4, we cultured naive lymph node follicular B cells of each genotype in LPS. Although all genotypes robustly proliferated, as assessed by CTV dilution, there was a decrease in cell yields without PU.1 and SpiB (Fig. 7e). LPS induced a strong

differentiation response to CD138$^+$ plasma cells in control B cells that was augmented as a proportion of live cells at day 4, in the PU1 SpiB DKO cells (Fig. 7e), in a manner similar to that observed for CD40 L + IL-4 cultures (Fig. 5a). However, after

adjustment for the different cell numbers obtained from the various genotypes, it was apparent that the total number of CD138$^+$ cells was reduced without PU.1 and SpiB (Fig. 7e). LPS-induced CSR to IgG3 was dependent on PU.1 (Fig. 7e), in a

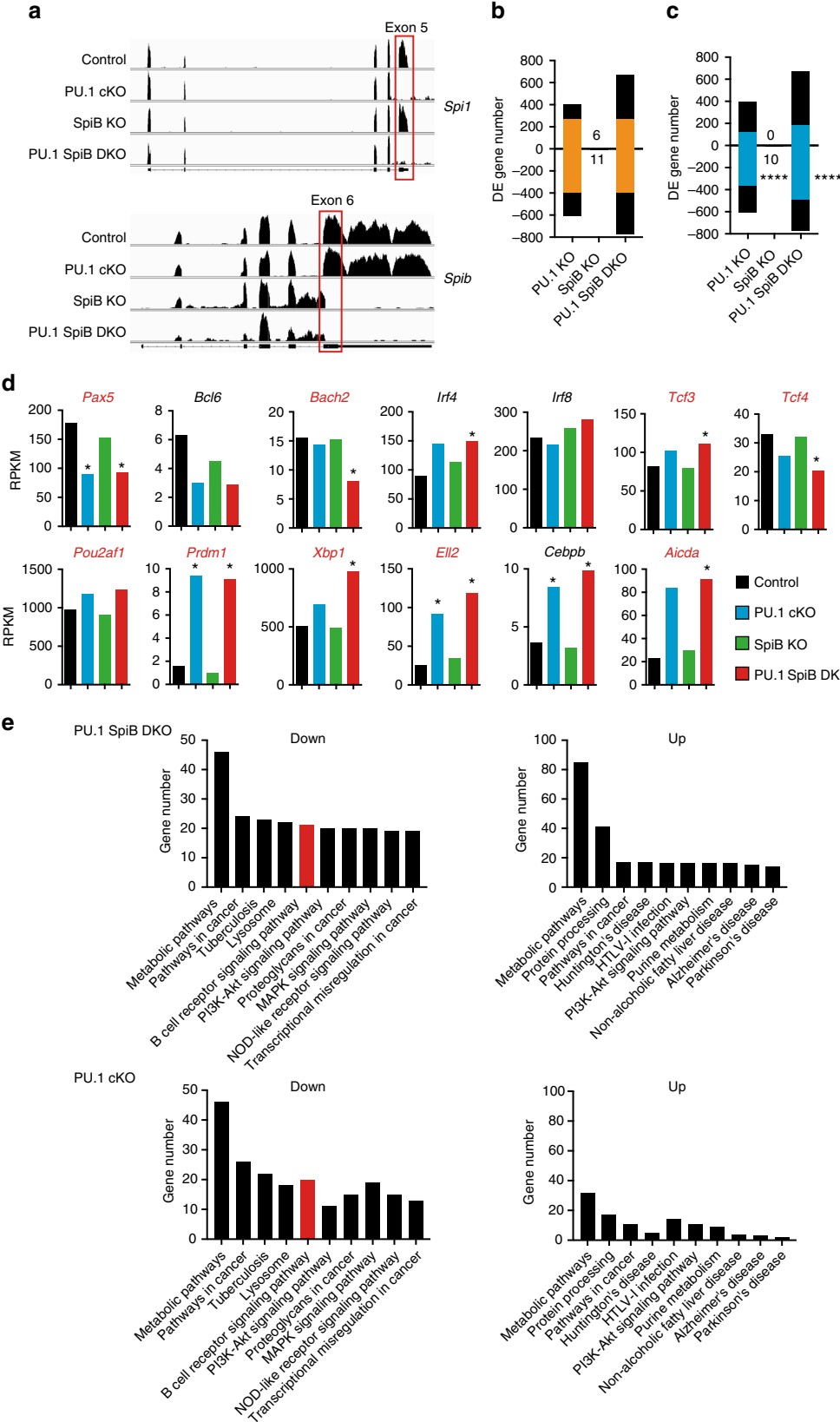

manner similar to what we observed for IgG1. Taken together these data demonstrate that PU.1 and SpiB redundantly control the receptors for several major pathways by which B cells sense and respond to environmental cues.

## Discussion

Humoral immunity is underpinned by the ability of B cells to sense and integrate signals from their environment and respond with the appropriate intrinsic differentiation program. These environmental cues include the cognate antigen, T cell help and survival factors such as BAFF. In this study, we show that the Ets family transcription factors, PU.1 and SpiB, well-known regulators of early lymphopoiesis, are redundantly required for the normal expression and function of virtually all pathways by which B cells sense and respond to the surrounding milieu. In contrast plasma cell development was either unaffected or accelerated by the loss of both factors. Thus PU.1 and SpiB act redundantly to balance the proliferative and differentiation response of B cells that is essential for protective humoral immunity.

Within the Ets family, PU.1 and SpiB are the genes most highly related to each other. They bind to the same consensus DNA recognition site, either on their own or in complex with IRF4 or IRF8[17, 18]. Both factors are constitutively expressed throughout B cell development, with SpiB expression being rapidly extinguished upon plasma cell differentiation, while PU.1 remains expressed at a low concentration and can, at least in culture, still occupy many thousands of binding sites in the genome[27, 30]. The similarity of this pair of transcription factors has long complicated the interpretation of their function in B cells. PU.1 null embryos die in late gestation and lack all lymphocytes[12], while removal of PU.1 in adult hematopoietic stem cells ablates B-cell development[9, 10]. Surprisingly, conditional inactivation of PU.1 in committed B cells using Cd19-Cre, which is first active during B cell development in the bone marrow, results in no pronounced defects in B cell production or function beyond a role in B1 cell formation[10, 13–16]. Loss of SpiB is also compatible with normal B cell development, however SpiB KO mice have defects in the GC reaction at later times points after immunization[22]. In this study, we observed normal numbers of peripheral B cell subsets in mice lacking PU.1 or SpiB, as well as functional GC, at the peak time post immunization (day 14). In contrast to our previous studies using Cd19-Cre[6, 14], we observed a partial impairment in GC responses in vivo and enhanced plasma cell differentiation in vitro in the absence of PU.1. The major difference between the studies is that here we have utilized Cd23-Cre that is active from the transitional stage in the spleen[21], sparing the bone marrow development process entirely. This, as discussed below may impact on the ability of SpiB to compensate for the loss of PU.1.

Previous attempts to investigate the degree of functional compensation between PU.1 and SpiB have been hampered by the complete absence of B cells in PU.1-deficient mice[12]. As a compromise, B cell development was examined in $Spi1^{+/-}Spib^{-/-}$ mice, which show a 9-fold decrease in splenic follicular B cells[19, 23]. The residual B cells in those mice, many of which were MZ B cells, appeared to be functionally defective, unable to contribute to GCs and impaired in their signaling responses through the BCR[19] and TLRs[31]. More recently, conditional inactivation of PU.1 on a SpiB KO background using Cd19-Cre, revealed a dramatic loss of follicular B cells and an expansion of the pre-B-cell compartment that ultimately generated B-ALL at high frequency[15, 32]. By utilizing Cd23-Cre our study has bypassed the important function of these proteins at the pre-B-cell stage and allowed the first robust analysis of the functionality of PU.1 SpiB DKO B cells. Although we found a 2-fold to 3-fold reduction in the frequency of follicular B cells in the DKO mice, these data should be interpreted cautiously, as we observed strong retention of the floxed exon 5 of Spi1 in B cells only in the absence of Spib, suggesting that these two proteins are redundantly essential for follicular B cell survival. Interestingly, the gene encoding the BAFF-R, Tnfrsf13c, is a direct PU.1/SpiB target gene providing a likely mechanism for the reduced follicular B cell numbers[33].

The redundancy between PU.1 and SpiB was also apparent on the molecular scale, with both the number of DE genes and the magnitude of change being increased in the PU.1 SpiB DKO activated B cells. Importantly, exon 5 of Spi1 was efficiently excised in the DKO B cells after in vitro activation for 48 h, allowing a full examination of the transcriptional consequences of gene inactivation at this stage. As with the cellular data, redundancy between PU.1 and SpiB was observed. This agrees well with ChIPseq studies that show that the vast majority of SpiB occupied sites in the B cell genome also bind PU.1[18]. Pathway analysis of these data revealed a striking role for PU.1 and SpiB in directly regulating the expression of many components of the BCR signaling pathway, including the previously characterized targets, Blnk[32] and Btk[34, 35]. Although the reduced expression of the individual target genes was in each case modest, combined reductions resulted in grossly impaired in vitro responsiveness to anti-IgM and a total ablation of the antigen-specific immune response. The function of PU.1 and SpiB was not restricted to the BCR pathway, as the ability of the double mutant follicular B cells to respond to antigen-independent simulation, either via T cells derived signals such as CD40 L or microbial stimuli such as LPS was also diminished, again likely due to the direct regulation of these key signaling pathways by PU.1 and SpiB.

In contrast to the essential function of PU.1 and SpiB to allow B cell responsiveness to environmental cues, both factors act as negative regulators of plasma cell development, in steady-state conditions in vivo and after activation in vitro. The increased

---

**Fig. 5** The transcriptional changes associated with the loss of PU.1 and SpiB. Resting lymph node B cells from mice of the indicated genotype were cultured in the presence of CD40 L + IL-4 for 48 h. RNA was extracted and whole transcriptome sequencing performed on duplicates of each cell population. **a** The read coverage showing deletion of exon 5 of Spi1 (PU.1) and exon 6 of Spib in the appropriate genotypes is shown mapped to the exon-intron structure. **b**, **c** The number of differentially expressed (DE) genes (defined as $\log_2$ fold change >0.6-fold, $\log_2$ RPKM (reads per kilobase of exon model per million mapped reads) >0, $P < 0.15$) identified for each genotype is shown relative to control B cells. Genes upregulated in the absence of the indicated transcription factor(s) are shown by bars above the line while genes down regulated are shown below. Values indicate the number of DE genes from SpiB KO cells. The identity of the DE genes is provided in Supplementary Data. **b** Orange bars show the number of DE genes common between the PU.1 cKO and PU.1 SpiB DKO genotypes. **c** Blue bars show the number of DE genes that have PU.1 peaks ($P < 10^{-6}$). P-values indicate the enrichment of PU.1 binding in down regulated DE genes compared to the total binding of PU.1 in non-DE genes using a Chi-square test. ****$P < 0.0001$. **d** The expression (in RPKM) of important regulators of mature B cell and antibody-secreting cell biology is shown for each of the four genotypes examined. *Indicates genes that are DE relative to controls (as defined in **a**). Gene names with PU.1 peaks ($P < 10^{-6}$) are indicated in red. **e** KEGG pathway analysis of genes regulated by PU.1 and PU.1/SpiB. Genes were first divided into those that were down or upregulated in the indicated genotypes. The number of genes that are associated with each statistically enriched ($P < 0.05$) functional category is shown

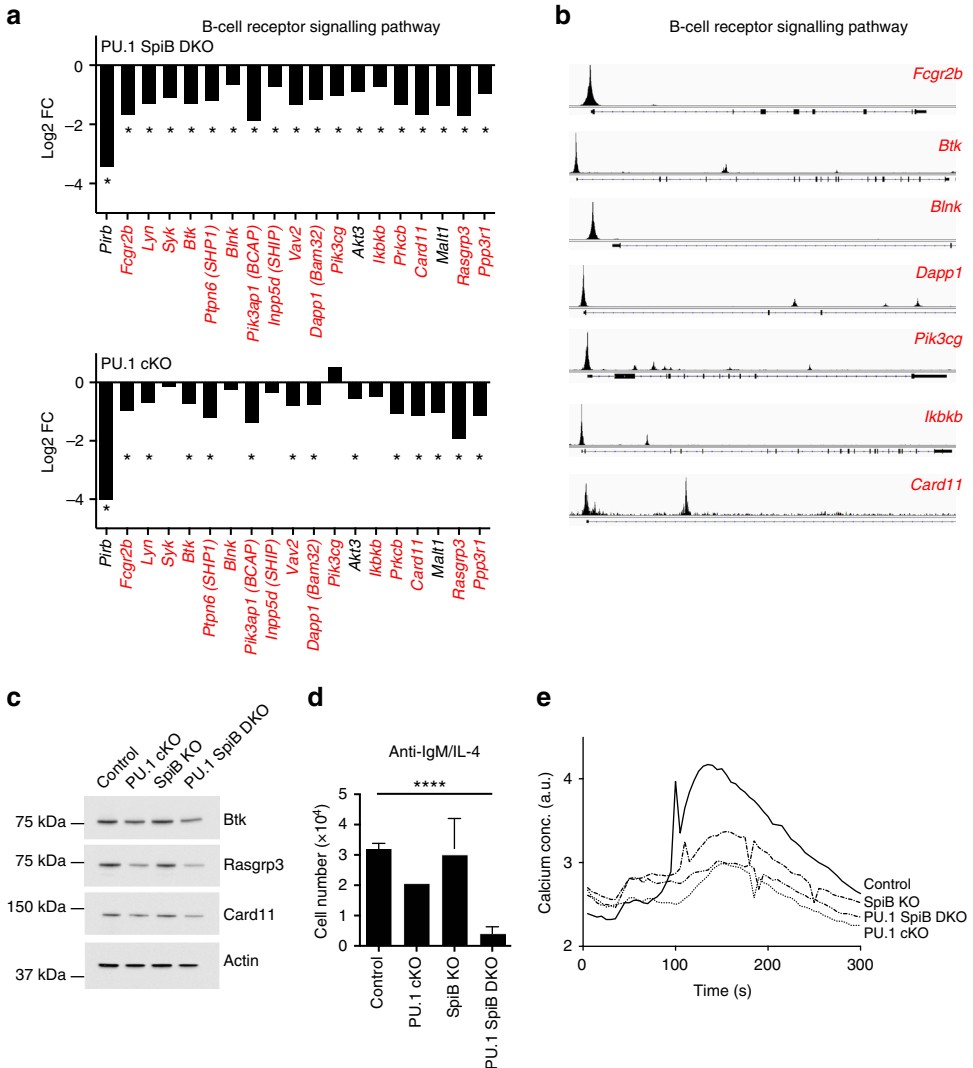

**Fig. 6** PU.1 and SpiB maintain expression of multiple BCR components. **a** Gene expression (from Fig. 5) for the indicated genotype is plotted as the mean fold change (log$_2$, from 2 experiments) relative to the expression in the control genotype. *Indicates genes that are DE relative to controls (defined as log$_2$ fold change >0.6-fold, log$_2$ RPKM (reads per kilobase of exon model per million mapped reads) >0, $P < 0.15$). **b** PU.1 ChIPseq identifies promoter-binding sites for PU.1 in many components of the BCR signaling pathway. Gene structures are shown below. Gene names in **a**, **b** with PU.1 peaks ($P < 10^{-6}$) are indicated in red. **c** Lysates were generated from activated B cells (cultured for 48 h in CD40 L + IL-4) of the indicated genotype. Western blotting was performed for the indicated proteins. Actin serves as a control for protein loading. Data in **b**, **c** are representative of two experiments each. **d** Loss of PU.1 and SpiB leads to decreased viability in response to BCR signaling. Resting lymph node B cells from mice of the indicated genotype were labeled with CTV and cultured in the presence of anti-IgM + IL-4 for 3d before flow cytometric analysis. The number of live cells is plotted for each genotype examined. Data are the mean ± s.d. for three independent experiments. $P$-values compare the indicated samples (two tailed $t$-test). ****$P < 0.0001$. **e** Resting B cells of the indicated genotype were cultured with CD40 L + IL-4 for 48 h before being washed and rested for one hour. Cells were then loaded with Indo-1 AM before basal fluorescence was measured for 30 s. Anti-IgM was then added and fluorescence changes recorded for an additional 2.5 min. Results are presented as arbitrary units of changes in fluorescence ratio over time. Data are representative of two experiments

plasma cell frequency in vivo was predominantly IgM secreting cells, with a milder increase in IgG3 production, isotypes that often derive from MZ B cells[24]. In keeping with this conclusion, MZ B cell frequency was unaltered in PU.1 SpiB DKO and mildly increased in $Spi1^{+/-}Spib^{-/-}$ spleens[23]. The preferential retention of MZ B cells is potentially also due to the role of PU.1 and SpiB in activating the expression of key components of the BCR pathway, as impaired BCR signaling is known to favor MZ over follicular B cell fate[36]. The splenic MZ B cells from $Spi1^{+/-}Spib^{-/-}$ mice have been shown to be impaired in their responses to several microbial products, including LPS, because of reduced $Nfkb1$ expression[31]. In contrast our RNA-seq and ChIPseq data derived from in vitro activated lymph node follicular B cells pointed to

defective $Tlr4$ expression as the underlying cause of the reduced responsiveness to LPS in PU.1 SpiB DKO B cells.

PU.1 cKO and PU.1 SpiB DKO B cells also produced markedly more plasma cells after in vitro activation with CD40 L + IL-4 or LPS + IL-4. This is in keeping with our recent report of increased plasma cell differentiation in the absence of PU.1 and IRF8[6] and occurred despite the lower overall cell yield without PU.1 and SpiB. PU.1/SpiB loss resulted in reduced expression of key B cell program regulators such as $Pax5$, $Bcl6$, and $Bach2$, while plasma cell promoting factors such as $Prdm1$, $Xbp1$, $Ell2$, and $Cebpb$ were increased in expression. As with PU.1/IRF8 DKO B cells[6], $Aicda$ expression was increased in the absence of PU.1 and SpiB suggesting that reduced expression of this enzyme was not the

underlying cause of the decreased CSR in PU.1 SpiB DKO B cells. Collectively these data suggest that PU.1 and SpiB act to balance B cell proliferation with the need to produce sufficient numbers of plasma cells, with PU.1 and SpiB deficiency tipping the balance away from clonal expansion and towards terminal differentiation.

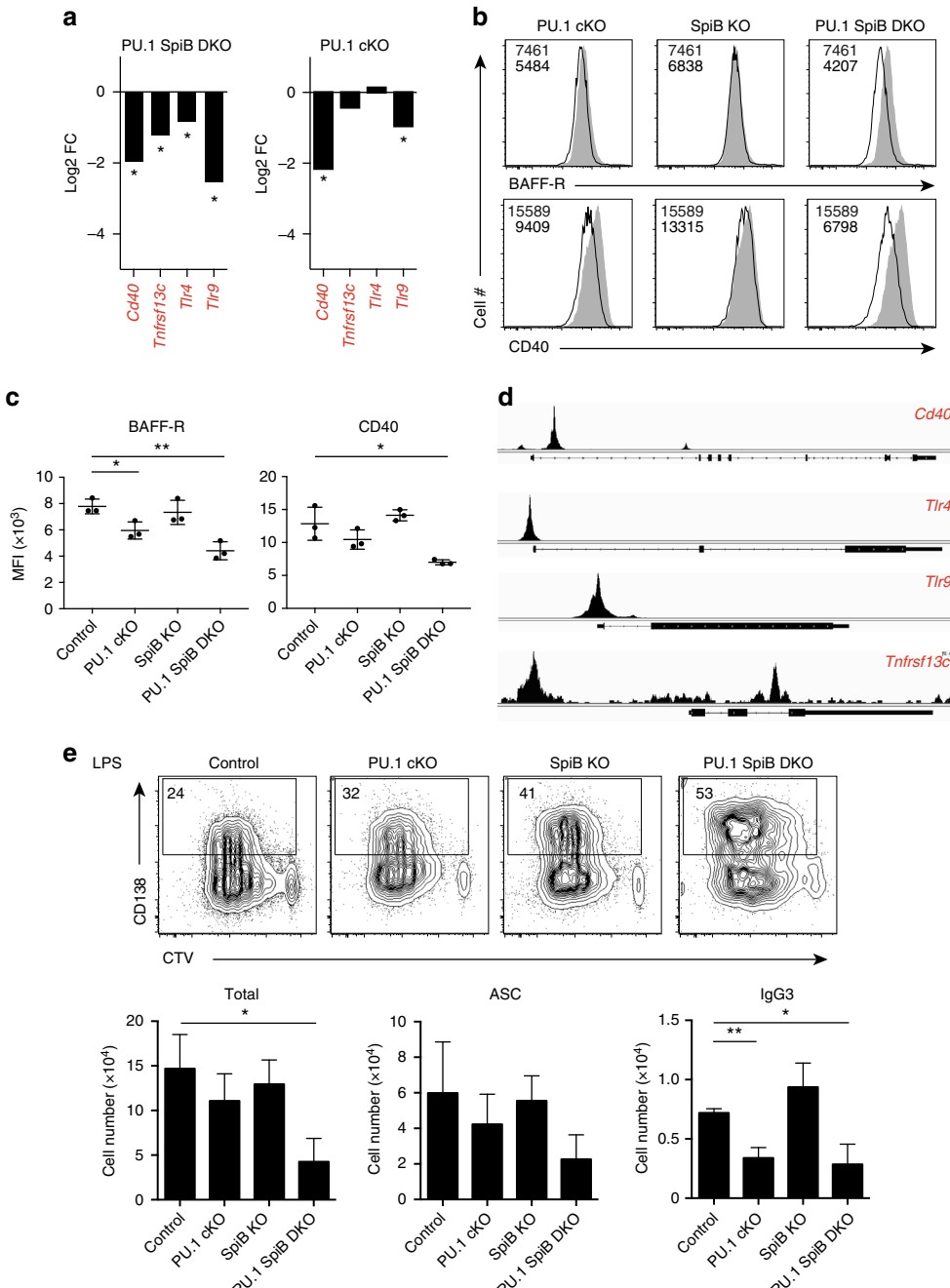

**Fig. 7** PU.1 and SpiB are required to maintain the expression of several immune sensors. **a** Gene expression (from Fig. 5) for the indicated genotype is plotted as the mean fold change (log$_2$, from 2 experiments) relative to the expression in the control genotype. *Indicates genes that are DE relative to controls (defined as log$_2$ fold change >0.6-fold, log$_2$ RPKM (reads per kilobase of exon model per million mapped reads) >0, $P < 0.15$). **b** Flow cytometric analysis showing the expression of BAFF-R or CD40 on activated B cells of the indicated genotype (black line), relative to control B cells (solid gray), 48 h after stimulation with CD40 L + IL-4. **c** Mean fluorescence intensity (MFI) is graphed as the mean ± s.d. Each circle represents the results from an individual mouse. $P$-values compare the indicated samples (two tailed $t$-test). *$P < 0.05$, **$P < 0.01$. Data in **b**, **c** are from 3 experiments. **d** ChIPseq identifies proximal promoter-binding sites for PU.1 for the genes shown in **a**. Gene structures are shown below. Gene names in **a**, **d** with PU.1 peaks ($P < 10^{-6}$) are indicated in red. Data are representative of two experiments. **e** Impaired survival of PU.1 SpiB DKO B cells in response to the TLR4 agonist LPS. Resting lymph node B cells from mice of the indicated genotype were labeled with CTV and cultured in the presence of LPS for 4d before flow cytometric analysis. Numbers in the boxes are the proportion of CD138$^+$ antibody-secreting cells (ASC). Data are representative of 3 experiments. Quantitation of the total cell number (left), total number of ASCs (middle) and IgG3$^+$ cells (right) is shown in the lower graphs. Data are the mean ± s.d. for three independent experiments. $P$-values compare the indicated samples (two tailed $t$-test). *$P < 0.05$, **$P < 0.01$. The full gating strategies are shown in Supplementary Fig. 9A and B, respectively

There is to date only limited understanding of the unique function(s) of SpiB, although it is of note that *Spib* expression is restricted to B cells and pDCs[37], lineages that have a distinctly low rate of PU.1 expression. Our finding of increased expression of genes associated with pDCs, macrophages and T cells in naive B cells lacking SpiB is reminiscent of similar observations of PU.1 repression of natural killer cell associated genes in pro-B[38] and pro-T[39] cells and suggests a previously unappreciated function of SpiB to repress lineage inappropriate gene expression in B cells that warrants closer examination in a future study. PU.1, in contrast, has a broad expression pattern and has been intensively studied in many blood cell lineages. One of the recurrent themes of PU.1 function in haematopoiesis is its pivotal role in activating the expression of a range of receptors, including the genes encoding MCSFR, GMCSFR, IL3R, IL7R, and Flt3 and many homing molecules such as integrins[8, 9, 40–43]. The data presented in this study provide a novel example of this paradigm, with PU.1 controlling the interplay between the extrinsic signals that control B cell homeostasis and responsiveness and intrinsic programs that govern differentiation events essential for humoral immunity.

## Methods

**Mice and immunizations**. The *Spi1*[fl/fl9], *Spib*[−/− 22], and *Cd23*-cre[21] mouse strains have been previously reported. All mice were maintained on a C57Bl/6 genetic background and analyzed between 6–20 weeks of age. Animals were aged and sex matched within individual experiments. The genotypes analyzed were control mice (*Cd23*[T/+]), PU.1 cKO mice (*Spi1*[fl/fl] *Cd23*[T/+]), SpiB KO mice (*Spi1*[fl/fl] *Cd23*[+/+] *Spib*[−/−]) and PU.1 SpiB DKO mice (*Spi1*[fl/fl] *Cd23*[T/+] *Spib*[−/−]). Mice were immunized at 6–20 weeks of age with 4 (hydroxy-3-nitrophenyl) acetyl (NP)-keyhole limpet hemocyanin (KLH) made at a molar ratio of 17:1 (NP/KLH). Antigen was precipitated in alum at a concentration of 1 mg/ml and delivered by i.p. injection (100 μg/mouse). Mice were sacrificed 14 d post immunization and single-cell suspensions were made from the spleen and bone marrow for analysis. All animal experiments were conducted according to the protocols approved by the Walter and Eliza Hall Institute animal ethics committee.

**Antibodies and flow cytometry**. Single-cell suspensions were stained with mAb against mouse B220 (RA3-6B2; BD Pharmingen; 1 : 200), IgM (331.12; 1 : 800), IgD (1126 C; 1 : 800), Gr-1 (RB6-8C5; 1 : 800), CD21 (7G6; 1 : 2000), CD23 (B3B4; Biolegend; 1 : 200), IgG1 (X56; BD Pharmingen; 1 : 300), IgG3 (pooled antisera; Southern Biotech; 1 : 250), CD138 (281-2; BD Pharmingen; 1 : 600), FcgR (2.4G2; 1 : 10), BAFF-R (7H22-E16; BD Biosciences; 1 : 100), CD40 (11-0402-86; eBioscience; 1 : 200) and CD98 (RL388; Biolegend; 1:500). Cells were analyzed on FACSCanto flow cytometers and cell sorting was carried out using FACSDiVa or Aria flow cytometers (BD Biosciences). Intracellular staining for BCL6 (7D1) was performed using the FoxP3 staining buffer set (eBiosciences). All Abs were produced in-house unless otherwise indicated. Antigen-specific B cells were identified by binding NP coupled to phycoerythrin.

**ELISpot and ELISA assays**. The frequency of NP-specific antibody-secreting cells was determined by ELISpot assay. $1 \times 10^6$ splenocytes were plated into replicate wells of 96-well cellulose ester–based plates (MAHA-S45-10; Millipore Corp.) coated with NP$_{17}$–BSA, followed by 20 h culture in RPMI medium containing 50 μM 2-mercaptoethanol and 5% fetal calf serum. Plates were washed, and bound NP-specific IgG1 was revealed with goat anti-mouse IgG1 conjugated to horseradish peroxidase (1070-05; Southern Biotechnology Associates; 1 : 500) and visualized by the addition of 3-amino-9-ethyl carbazole. Spots, each representing a single antibody-secreting cell, were counted using an ELIspot reader (Autoimmune Diagnostika).

Serum Ig concentrations were measured by ELISA. All antibodies were from Southern Biotechnology Associates. Plates were coated with either goat anti-mouse IgM (1020-01; 1 : 500), IgA (1040-01; 1 : 100), IgG1 (1070-01; 1 : 500), IgG2b (1090-01; 1 : 500), IgG2c (1080-01; 1 : 500) or IgG3 (1100-01; 1 : 500), washed and exposed to a 1 : 200 dilution of mouse serum. Plates were washed and the bound Ig detected using horseradish peroxidase conjugated goat anti-mouse IgM (1020-05; 1 : 500), IgG1 (1070-05; 1 : 500), IgG2b (1090-05; 1 : 500), IgG2c (1080-05; 1 : 500), IgG3 (1100-05; 1 : 500) or biotin conjugated goat anti-mouse IgA (1040-08; 1 : 1000) followed by streptavidin- horseradish peroxidase (7100-05, 1 : 1000). Plates were visualized by the addition of 2,2′-azino-bis(3-ethylbenzothiazoline-6-sulphonic acid). Purified Ig isotype standards were used to determine the serum Ig concentrations.

**Immunofluorescence**. Spleens were embedded in OCT (Tissue Tek, Sakura) and snap frozen on dry ice in a bath of isopentane. 10 μm-thick sections were fixed and permeabilized in cold acetone for 10 min, then saturated with 20% BSA in PBS for 15 min. Sections were stained using the following antibodies: B220 (RA3-6B2, FITC; 1 : 200), CD3 (2C11; produced in house; 1 : 300), IgM (donkey polyclonal, AlexaFluor 594, Jackson Immunoresearch; 1 : 800), IgD (1126 C, AlexaFluor 647, BD pharmingen; 1 : 100), goat anti-hamster IgG (DyLight 594 or 649; Biolegend; 1 : 200). Slides were mounted in Mowiol (Sigma-Aldrich) containing 2.5% DABCO (Sigma). Images were acquired using a Zeiss LSM780 confocal microscope, and analyzed with the Fiji software.

**Cell culture**. Naive lymph node B cells were purified by negative selection using the B cell Isolation kit (130-090-862; Miltenyi Biotec). Cultures were seeded at $1–2 \times 10^5$/ml with optimal concentrations of recombinant CD40 L (1163-CL; R&D Systems; 50 ng/ml) and IL-4 (404-ML; R&D Systems; 10 ng/ml,), or LPS (L3024; Sigma-Aldrich; 20μg/ml) ±IL-4 for 4 days before analysis or with anti-IgM (115-006-075; Jackson Immunoresearch; 10 μg/ml) and IL-4 (as above) for 3 days. For cell division tracking, purified B cells were labeled with Cell Trace Violet (CTV, Molecular Probes), prior to cell culture.

**Calcium flux**. Naive resting B cells were isolated as described above and cultured for 48 h in CD40 L + IL-4, to ensure complete deletion of the *Spi1* allele in PU.1 SpiB DKO B cells. Cells were then washed extensively and rested for one hour before being resuspended in loading buffer (KDS-BSS buffer containing 0.1% glucose and 0.1% FCS) containing Indo-1 acetoxymethyl (Molecular Probes). Following an incubation for 40 min at 37 °C, loaded cells were washed extensively and then resuspended in the loading buffer indicated above. Calcium flux was measured by flow cytometry. Baseline fluorescence was established for 30 s before anti-IgM (10 μg/ml; Jackson Immunoresearch) was added and the subsequent calcium flux followed for the indicated times.

**Whole transcriptome analysis**. RNA was isolated from either naive or activated (48 h in CD40 L + IL-4 or LPS + IL-4) lymph node derived B cells using the RNeasy Plus Mini kit (74,134; Qiagen). Two biological replicates were generated and sequenced for each sample. mRNA was subjected to transcriptome re-sequencing on an Illumina NextSeq sequencer. Between 15 and 90 million reads were analyzed per sample. Each sample's read set was aligned to the GRCm38/mm10 build of the *Mus musculus* genome using the Subread aligner[44]. Only uniquely mapped reads were retained. Genewise counts were obtained using featureCounts[45]. Reads overlapping exons in annotation build 38.1 of NCBI RefSeq database were included. Genes were filtered from downstream analysis if they failed to achieve a CPM (counts per million mapped reads) value of at least 0.5 in at least two libraries. Counts were converted to $\log_2$ counts per million, quantile normalized and precision weighted with the "voom" function of the limma package[46, 47]. A linear model was fitted to each gene, and empirical Bayes moderated t-statistics were used to assess differences in expression[48]. Genes were called differentially expressed if they achieved a false discovery rate of 0.15 or less with a fold change >1.5. Pathway analysis was performed using the KEGG (Kyoto Encyclopedia of Genes and Genomes) database (http://www.genome.jp/kegg/)[49] while analysis for association with particular cell lineages was performed with Enrichr[50, 51].

**PU.1 binding analysis**. The PU.1 ChIPseq dataset from LPS activated plasmablasts was downloaded from the Gene Expression Omnibus (GEO) database (GSM1843342). Sequence reads were mapped to mouse genome mm10 using Subread aligner. Only uniquely mapped reads were kept. PU.1 binding peaks were called by using MACS2 program with a false discovery rate cutoff of $<10^{-6}$. Called peaks must also have at least 200 support reads in the 200 bp region centered on the binding summit. Peaks were assigned to genes if they occurred <20 kb upstream of the transcriptional start sites, within the gene body an <5 kb downstream of the end of transcription. Within these parameters multiple gene assignment was allowed.

**Western analysis**. The polyclonal antibodies against Btk (D3H5; 1 : 1000), Rasgrp3 (C33A3; 1 : 1000) and Card11 (1D12; 1 : 1000) were from Cell Signaling Technology while the antibody against actin (I-19; 1 : 3000) was from Santa Cruz Biotechnology. Protein extracts corresponding to equal cell numbers were loaded onto the gel and subjected to western blotting using standard techniques.

**Statistical analysis**. Statistical significance of non RNA-Seq data was assessed with Prism6 (GraphPad). t tests for all single comparisons were unpaired, assumed Gaussian distribution and that both populations have the same standard deviation (SD). Enrichment of PU.1 binding in DE gene lists was determined using $\chi^2$-tests. Bar graphs display the arithmetic mean ± s.d. unless otherwise stated.

**Data availability**. Sequence data that support the findings of this study have been deposited in GEO with the primary accession code GSE90094.

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

## Acknowledgements

We thank Patrick Leung and Nadia Iannarella for technical assistance and Meinrad Busslinger (IMP Vienna) for mice. This work was supported by a National Health and Medical Research Council (NHMRC) of Australia grants (1054925 to D.M.T. and S.L.N., 1058238 to S.L.N. and 1060675 to D.M.T.), Victorian State Government Operational Infrastructure Support, Australian Government NHMRC IRIIS, Lupus Research Alliance

(to D.M.T.) and the Multiple Myeloma Research Foundation (MMRF, to S.L.N. and J.T.). S.N.W. was supported by a Walter and Eliza Hall Trust Centenary Fellowship, W.S. by a Commonwealth Serum Laboratories Centenary Fellowship.

## Author contributions

S.N.W. performed most of the experiments; J.T. provided the immunofluorescence; S.T., A.L., O.K. assisted in some experiments; Y.L. and W.S. performed the bioinformatics analysis; L.A.G.-S. provided a key reagent; D.M.T. and S.L.N. supervised the research; S.N.W. and S.L.N. designed the experiments, analyzed the data and wrote the manuscript.

## Additional information

**Competing interests:** The authors declare no competing financial interests.

