## [Peer review file · Nature Communications]

Reviewers' comments:

Reviewer #1 (Remarks to the Author):

By using CD23-cre to delete PU.1 in SpiB knockout mice so as to bypass the well-established block of B cell development due to combined loss of PU.1 and SpiB, Willis and colleagues have examined functions of these two redundant Ets family transcription factors in regulating physiology of mature B cells at the steady state and following immunization. Previous work from this group has shown PU.1 restrains plasma cell development in response to immune challenge. In this study, these authors further report a collection of phenotypes of the double KO mice, ranging from follicular B cell maintenance/survival, steady-state plasma cell development to GC development following active immunization, which in combination suggest PU.1/SpiB regulate multiple aspects of mature B cell biology. In correlation with these observations, by ChIP data mining and RNA-seq analyses, the authors provide evidence that PU.1/SpiB control expression of surface receptors for microbial stimuli, antigen, and T cell-derived signals.

Whereas the study is technically well performed and reasonably interpreted, it overall represents only incremental advance in our understanding of these important transcription factors. As cited by the authors, numerous earlier studies of PU.1 +/- SpiB -/- mice have revealed defects of the knockout B cells in responding to BCR and TLR stimulation (e.g. ref. 18, 19, 31). In fact, most of functional phenotypes reported here have been seen before in association with one or another form of combined PU.1/SpiB deficiencies, and the new addition here really rests on the use of CD23-cre to re-confirm those phenotypes. The RNA-seq data are interesting but do not offer firm mechanistic insights. It is useful to define the degree of redundancy between PU.1 and SpiB in regulating the various B cell functions, as attempted here, but no major conceptual advances emerge.

Minor points:

Have the authors attempted to validate their RNA-seq data by qPCR and to examine protein expression? In the earlier report by Garrett-Sinha et al (ref 19), expression of signaling components downstream of BCR were found similar between WT and PU.1 +/- SpiB -/- cells, even though the two types of cells differ much in signaling competence. It is possible that small differences in many components along the same signaling pathway collectively give rise to a profound overall defect.

Reviewer #2 (Remarks to the Author):

This manuscript provides a comprehensive analysis of the control of mature B cells by the related transcription factors PU.1 and SpiB. The results extend previous investigations in this area and provide new and interesting findings concerning the regulation of plasma cell differentiation and the transcription of the genes encoding key B cell surface receptors by these transcription factors. This is an important contribution to the field and provides a detailed and insightful picture of molecular redundancy. The quality of the data is outstanding and it is described and interpreted clearly and carefully.

My only other comment would be that confirmatory flow cytometry plots showing reduced expression of cell surface receptors BAFFR, CD40 and potentially TLR4 would help bolster the authors' conclusions concerning B cell responsiveness and survival in the absence of PU.1 and SpiB.

Reviewer #3 (Remarks to the Author):

In this manuscript Willis et al. present an interesting study of the joint role of ETS-family transcription factors PU.1 and SPIB in mature B-cell function and differentiation. These transcription factors play partially redundant roles, and are essential for B-cell development. Prior similar models relying on conditional deletion of PU.1 in a SPIB deficient background using CD19-Cre demonstrated a critical requirement during B-cell ontogeny leading ultimately to development of B-ALL. Here the authors have used the CD23-Cre model, to allow deletion of PU.1 in SPIB deficient background at the mature B-cell stage, circumventing the developmental block observed in the CD19-Cre model. The experiments demonstrate that PU.1 and SPIB are together essential for the establishment of normal follicular B-cells, for the establishment of efficient germinal centre responses, and for normal signaling via the B-cell receptor complex in particular. By integrating RNA-seq based gene expression in the different genotypes with pre-existing ChIP-seq data the authors seek to differentiate the relative contribution of PU.1 and SPIB to the phenotypes observed in vivo.

The manuscript provides valuable and novel insight into the role of these partially redundant transcription factors in B-cell differentiation, in particular in relation to plasma cell and germinal center responses. However a major limitation stems from the apparent discrepancy between the in vivo and in vitro phenotypes, and from the fact that selective pressure to retain a functional copy of PU.1 leads to incomplete inactivation of the gene in follicular B-cells necessitating in vitro approaches to explore gene expression effects which then have significant caveats.

Specific comments:

1) The apparent discrepancy between in vivo and in vitro phenotypes revolves around the in vivo results obtained for SPIB inactivation and the eventual conclusion that PU.1 provides the dominant effects on gene regulation and function when both PU.1 and SPIB are inactivated. A range of in vivo and in vitro observations across the four genotypes are juxtaposed by a set of in vitro data using a particular stimulation condition which show that after CD40 and IL4 mediated stimulation for 48 hours, only 8 genes are differentially expressed in the absence of SPIB, while hundreds are differentially expressed in the PU.1 CKO setting or in the double SPIB PU.1 deficient setting (Figure 5). Yet this result is set against a range of data that support a quite profound effect for SPIB deficiency alone, suggesting that the experimental system on which the RNA-seq data is based is providing an incomplete picture. Consider that: in Figure 1A loss of SPIB is associated with a greater loss of CD23 expression intensity (which may in itself account for the partial retention of PU.1 in the SPIB deficient background), and at the same time a greater increase in MZ B-cells in Figure 1B; in Figure 2 SPIB deficiency leads to the greater increase in plasma cell numbers in spleen and bone marrow; and in Figure 3 SPIB deficiency results in the more significant changes in immunoglobulin isotypes; in Figure 7D SPIB deficiency leads to a substantial decrease in BCR mediated calcium signaling, and appears to have the more substantial effect on LPS mediated B-cell proliferation in Figure 8C. Yet none of this is readily explained by the small number of differentially expressed genes identified in the experiments in Figure 5.

2) Considering the gene expression experiment itself in Figure 5, which provides the core of the mechanistic analysis, SPIB is inactivated in the germline while the PU.1 deletion occurs at the FO B-cell stage during an activation process in which SPIB, but not PU.1, is repressed, thus providing less chance for compensatory changes. Secondly and more significantly as shown in Figure 4 under the conditions used in the absence of PU.1 there is quite a profound acceleration in plasma cell differentiation by day 4 (consistent with the authors previous work showing that PU.1 functions as a repressor of plasma cell differentiation in conjunction with IRF8, in a complex which is dependent on PU.1). Data for the relative phenotype at the 48h time point is not provided, but it is likely that differentiation will have been accelerated at this point as well. The large number of differentially expressed genes is thus likely to substantially derive from the enhanced differentiation process, which would be consistent with the gene functional enrichments observed. Thus this experiment does not allow the distinction between genes primarily regulated by PU.1, and genes secondarily regulated by the plasma cell differentiation program preferentially accelerated in the absence of PU.1 under the specific stimulation conditions used.

3) The need for evaluating gene expression at 48h of activation is driven by the retention of a PU.1

allele in the double deficient setting. But a comparison between PU.1 and SPIB single knockouts albeit not both in the same conditional manner would be possible using unstimulated FO B-cells. This would exclude effects of enhanced differentiation in one relative to the other population in the analysis of direct contributions to gene regulation. Equally taking additional earlier or later time point might help address the issue of different rates of differentiation as noted above.

4) In Figure 8C under stimuli of TLR mediated differentiation the phenotypes of the four different genotypes also appear more comparable than in the context of CD40 and IL4 stimulation. To confirm that the apparent selective effect of PU.1 on gene expression was not limited to the single stimulus condition of CD40 and IL4, it would be valuable to address differential gene expression in the context of TLR driven activation/differentiation at a similar time point. This could help substantiate the conclusion that the role of PU.1 is general and not stimulus specific.

5) In Figure 7 there is a link drawn between differential gene expression and impaired BCR signaling with loss of expression of many components in the PU.1 and PU.1/SPIB double deficient setting. In Figure 7D the SPIB deficient context is described as showing a "modest" reduction in BCR calcium flux with a "severe reduction in the absence of PU.1 or PU.1 and SPIB" (page 12 line 259-260), and yet this interpretation is not immediately obvious from the data where the major difference appears to be between wild type and any of the other genotypes. One might equally interpret the data shown as supporting a major difference between any of the knockout genotypes and wild type and a modest difference between the SPIB deficient and PU.1 or PU.1/SPIB double deficient cells. In order to relate the BCR signaling defect to the differential gene expression it would therefore be necessary to provide an explanation for the defect in the SPIB deficient context. Furthermore it would be important to consider the relative phenotypes of the cells responding to BCR ligation, and any impact of accelerated differentiation at the 48h time point of CD40 and IL4 stimulus used (as the same issue could apply as noted above).

Reviewer #1

By using CD23-cre to delete PU.1 in SpiB knockout mice so as to bypass the well-established block of B cell development due to combined loss of PU.1 and SpiB, Willis and colleagues have examined functions of these two redundant Ets family transcription factors in regulating physiology of mature B cells at the steady state and following immunization. Previous work from this group has shown PU.1 restrains plasma cell development in response to immune challenge. In this study, these authors further report a collection of phenotypes of the double KO mice, ranging from follicular B cell maintenance/survival, steady-state plasma cell development to GC development following active immunization, which in combination suggest PU.1/SpiB regulate multiple aspects of mature B cell biology. In correlation with these observations, by ChIP data mining and RNA-seq analyses, the authors provide evidence that PU.1/SpiB control expression of surface receptors for microbial stimuli, antigen, and T cell-derived signals.

Whereas the study is technically well performed and reasonably interpreted, it overall represents only incremental advance in our understanding of these important transcription factors. As cited by the authors, numerous earlier studies of PU.1^{+/}-SpiB^{-/-} mice have revealed defects of the knockout B cells in responding to BCR and TLR stimulation (e.g. ref. 18, 19, 31). In fact, most of functional phenotypes reported here have been seen before in association with one or another form of combined PU.1/SpiB deficiencies, and the new addition here really rests on the use of CD23-cre to re-confirm those phenotypes. The RNA-seq data are interesting but do not offer firm mechanistic insights. It is useful to define the degree of redundancy between PU.1 and SpiB in regulating the various B cell functions, as attempted here, but no major conceptual advances emerge.

RESPONSE: Although we acknowledge that reviewer 1 is correct that earlier studies of PU.1^{+/}-SpiB^{-/-} mice (as cited by us) have shown roles for these factors in B cell functions, very few mature B cells were produced in those mice, making the interpretation of any functional defects problematic. Our current studies have used a far superior model to examine the mature B cell specific function(s) of PU.1 and SpiB, providing robust conclusions that could not be deduced from the aberrant cells found in PU.1^{+/}-SpiB^{-/-} mice. For example, we show a very clear block in GC formation and a striking hyper plasma cell differentiation phenotype. We also have provided side-by-side comparisons of the PU.1 and SpiB single mutants, which enables us to deduce the level of redundancy in these proteins.

In terms of mechanistic insight, this study is the first to examine the transcriptomes of SpiB KO and PU.1/SpiB DKO mature naïve and activated B cells. This analysis and the validation with our previously reported PU.1 ChIPseq data identified a number of clear defects in environmental sensing by B cells that form the basis of our mechanistic conclusions and provides a new paradigm for the function of PU.1/SpiB in B cells. We believe these are important findings needed to properly place PU.1 and SpiB in the gene regulatory network controlling late B cell differentiation.

Minor points:

Have the authors attempted to validate their RNA-seq data by qPCR and to examine protein expression?

RESPONSE: This issue was also raised by Reviewer 2 and the editor, while Reviewer 3 raised a related point about PU.1/SpiB function in B cell cultures stimulated TLR ligands. To address all these issues together we have measured the protein expression of several of the key targets by flow cytometry, including CD40, BAFFR (Fig. 7D, S5) and CD23 (Fig. S1) or by Western blotting for Btk, Rasgrp3 and Card11 (Fig. 6C). Finally, we have now performed RNAseq on B cell activated by the combination LPS+IL-4 (in biological duplicate) as an independent validation on the RNA level and found similar changes in CD40, BAFFR, TLR4/9 and several BCR signaling components (Fig. S7).

In the earlier report by Garrett-Sinha et al (ref 19), expression of signaling components downstream of BCR were found similar between WT and PU.1+/-SpiB-/- cells, even though the two types of cells differ much in signaling competence. It is possible that small differences in many components along the same signaling pathway collectively give rise to a profound overall defect.

RESPONSE: We agree with the reviewer's interpretation of the data. The fold reduction in the expression of most individual signaling components and receptors is relatively small (although in all cases statistically significant) and it is the cumulative effect of these changes that we believe leads to the functional defects we observed. This interpretation agrees with our overarching conclusion that PU.1 and SpiB collectively control the environmental responsiveness of B cells through multiple mechanisms.

Reviewer #2

This manuscript provides a comprehensive analysis of the control of mature B cells by the related transcription factors PU.1 and SpiB. The results extend previous investigations in this area and provide new and interesting findings concerning the regulation of plasma cell differentiation and the transcription of the genes encoding key B cell surface receptors by these transcription factors. This is an important contribution to the field and provides a detailed and insightful picture of molecular redundancy. The quality of the data is outstanding and it is described and interpreted clearly and carefully. My only other comment would be that confirmatory flow cytometry plots showing reduced expression of cell surface receptors BAFFR, CD40 and potentially TLR4 would help bolster the authors' conclusions concerning B cell responsiveness and survival in the absence of PU.1 and SpiB.

RESPONSE: As outlined in our response to Reviewer 1, we have measured the protein expression of several of the key targets by flow cytometry, including CD40, BAFFR (Fig. 7D, S5) and CD23 (Fig. S1) or by Western blotting for Btk, Rasgrp3 and Card11 (Fig. 6C). Finally, we have now performed RNAseq on B cells activated by the combination LPS+IL-4 (in biological duplicate) as an independent validation on the RNA level and found similar changes in CD40, BAFFR, TLR4/9 and several BCR signaling components.

Reviewer #3

In this manuscript Willis et al. present an interesting study of the joint role of ETS-family transcription factors PU.1 and SPIB in mature B-cell function and differentiation. These transcription factors play partially redundant roles, and are essential for B-cell development. Prior similar models relying on conditional deletion of PU.1 in a SPIB deficient background using CD19-Cre demonstrated a critical requirement during B-cell ontogeny leading ultimately to development of B-ALL. Here the authors have used the CD23-Cre model, to allow deletion of PU.1 in SPIB deficient background at the mature B-cell stage, circumventing the developmental block observed in the CD19-Cre model. The experiments demonstrate that PU.1 and SPIB are together essential for the establishment of normal follicular B-cells, for the establishment of efficient germinal centre responses, and for normal signaling via the B-cell receptor complex in particular. By integrating RNA-seq based gene expression in the different genotypes with pre-existing ChIP-seq data the authors seek to differentiate the relative contribution of PU.1 and SPIB to the phenotypes observed in vivo.

The manuscript provides valuable and novel insight into the role of these partially redundant transcription factors in B-cell differentiation, in particular in relation to plasma cell and germinal center responses. However, a major limitation stems from the apparent discrepancy between the in vivo and in vitro phenotypes, and from the fact that selective pressure to retain a functional copy of PU.1 leads to incomplete inactivation of the gene in follicular B-cells necessitating in vitro approaches to explore gene expression effects which then have significant caveats.

Specific comments:

1) The apparent discrepancy between in vivo and in vitro phenotypes revolves around the in vivo results obtained for SPIB inactivation and the eventual conclusion that PU.1 provides the dominant effects on gene regulation and function when both PU.1 and SPIB are inactivated.

RESPONSE: Although we will respond to the reviewer's individual comments below, we would like to point out that our overarching conclusion is that PU.1 and SpiB act redundantly in B cell activation and differentiation. PU.1 individually regulated more genes than SpiB in both naïve and activated B cells, however the loss of both proteins resulted in considerably more gene expression changes and all the major phenotypes. Despite this redundancy distinct individual functions of PU.1 and SpiB were observed in this and previous studies.

2) A range of in vivo and in vitro observations across the four genotypes are juxtaposed by a set of in vitro data using a particular stimulation condition which show that after CD40 and IL4 mediated stimulation for 48 hours, only 8 genes are differentially expressed in the absence of SPIB, while hundreds are differentially expressed in the PU.1 CKO setting or in the double SPIB PU.1 deficient setting (Figure 5). Yet this result is set against a range of data that support a quite **profound** effect for SPIB deficiency alone,

suggesting that the experimental system on which the RNA-seq data is based is providing an incomplete picture.

RESPONSE: We previously focused our mechanistic studies on the activated B cells, as in that circumstance PU.1 was efficiently inactivated in the DKO. We did at the same time analyze naïve B cells from the 4 genotypes. This data is now shown in Fig. S4 and S5. Differential expression (DE) analysis revealed that PU.1 and SpiB regulated 547 and 332 genes respectively in naïve follicular B cells, with a clear redundancy observed in the absence of SpiB and a partial deletion of PU.1 (1306 genes). Interestingly, although most genes activated by PU.1 or SpiB were associated with the follicular B cell lineage, many of the SpiB regulated genes were repressed and were usually associated with other cell lineages including plasmacytoid DC, macrophages and CD8 T cells (Fig S4C). We intend follow up on this observation in the future. We hope that these additional experiments further clarify the individual functions of PU.1 and SpiB in mature B cells.

3) Consider that: in Figure 1A loss of SPIB is associated with a greater loss of CD23 expression intensity (which may in itself account for the partial retention of PU.1 in the SPIB deficient background), and at the same time a greater increase in MZ B-cells in Figure 1B;

RESPONSE: The reviewer is correct in concluding that SpiB plays a role in regulating CD23 (Fig. 1A and new S1), however the decrease is much more pronounced in the PU.1/SpiB DKO. We agree that the decreased CD23 potentially accounts for the inefficient PU.1 inactivation by CD23-Cre (this is a BAC-Tg), however we favor a functional selection against loss of both PU.1/SpiB, as the same naïve B cells show complete inactivation of PU.1 after 2 days in culture (Fig. 5A and S6C) despite CD23 expression being generally downregulated with differentiation.

The increase in MZ B cells in the absence of SpiB or PU.1/SpiB was variable and did not reach statistical significance despite us analyzing 7-10 mice/genotype.

4) in Figure 2 SPIB deficiency leads to the greater increase in plasma cell numbers in spleen and bone marrow;

RESPONSE: Again, the frequencies of plasma cells in the SpiB KO or PU.1/SpiB DKO were variable. Plasma cells were significantly increased in the PU.1/SpiB DKO in the BM and spleen, however SpiB KO showed an increase only in the spleen. While we agree with the reviewer that SpiB deficiency appears to enhance plasma cell differentiation in vivo (a novel observation), the increased phenotype observed in the DKO demonstrates functional redundancy, which is the major point we are trying to make with this figure. We have amended the results section to acknowledge better the individual function of SpiB in controlling plasma cell numbers.

5) and in Figure 3 SPIB deficiency results in the more significant changes in immunoglobulin isotypes

RESPONSE: We agree with the reviewer here and have modified the results section to reflect this.

6) ; in Figure 7D SPIB deficiency leads to a substantial decrease in BCR mediated calcium signaling,

RESPONSE: We agree with the reviewer that SpiB KO have impaired BCR signaling, although possibly a less severe impact than either PU.1 KO or PU.1/SpiB DKO (now Fig. 6E). However, it is striking that only the combined loss of PU.1/SpiB resulted in impaired cellularity after culture in anti-IgM/IL-4, which we again interpret as indicating evidence of functional redundancy between PU.1 and SpiB.

7) and appears to have the more substantial effect on LPS mediated B-cell proliferation in Figure 8C.

RESPONSE: We disagree with the reviewer here. Our FACS data for the proportion of CD138+ ASCs elicited by LPS shows a modest increase in frequency in PU.1 or SpiB KO and a further increase in DKOs. However, when we take into account the cellularity, it is clear that total cell number (and thus ASC number) is normal in the single KOs and specifically decreased in the PU.1/SpiB DKO.

8) Yet none of this is readily explained by the small number of differentially expressed genes identified in the experiments in Figure 5.

9) Considering the gene expression experiment itself in Figure 5, which provides the core of the mechanistic analysis, SPIB is inactivated in the germline while the PU.1 deletion occurs at the FO B-cell stage during an activation process in which SPIB, but not PU.1, is repressed, thus providing less chance for compensatory changes.

RESPONSE TO POINT 8 AND 9: We have addressed these questions by performing RNAseq in naïve Follicular B cells from the LNs of the 4 genotypes. See our response to point #2 above for the details and our interpretations.

Secondly and more significantly as shown in Figure 4 under the conditions used in the absence of PU.1 there is quite a profound acceleration in plasma cell differentiation by day 4 (consistent with the authors previous work showing that PU.1 functions as a repressor of plasma cell differentiation in conjunction with IRF8, in a complex which is dependent on PU.1). Data for the relative phenotype at the 48h time point is not provided, but it is likely that differentiation will have been accelerated at this point as well. The large number of differentially expressed genes is thus likely to substantially derive from the enhanced differentiation process, which would be consistent with the gene functional enrichments observed. Thus, this experiment does not allow the distinction between genes primarily regulated by PU.1, and genes secondarily regulated by the plasma cell differentiation program preferentially accelerated in the absence of PU.1 under the specific stimulation conditions used.

RESPONSE: The reviewer raises an important point that we have controlled for. Fig. S3 shows analysis of B220/CD138 and cell proliferation (assessed by CTV dilution) for all genotypes at 48hrs. These data show virtually no CD138+ ASCs at this time point, a finding compatible with our previous studies on PU.1/Irf8 DKO (Carotta JEM 2014) and that all cells have undergone 1-3 divisions. Our interpretation of both the current results and those of Carotta et al is that PU.1/SpiB controls the probability of differentiation of an activated B cell per cell division and not the rate of differentiation as measured by time.

10) The need for evaluating gene expression at 48h of activation is driven by the retention of a PU.1 allele in the double deficient setting. But a comparison between PU.1 and SPIB single knockouts albeit not both in the same conditional manner would be possible using unstimulated FO B-cells. This would exclude effects of enhanced differentiation in one relative to the other population in the analysis of direct contributions to gene regulation. Equally taking additional earlier or later time point might help address the issue of different rates of differentiation as noted above.

RESPONSE: We have addressed these questions by performing RNAseq in naïve Follicular B cells from the LNs of the 4 genotypes. See our response to point #2 above for the details and our interpretations. The caveat to this experiment is the incomplete deletion of PU.1 in the DKO cells, however as shown in Fig. S4, we still observed marked changes in gene expression in the PU.1/SpiB DKO genotype.

11) In Figure 8C under stimuli of TLR mediated differentiation the phenotypes of the four different genotypes also appear more comparable than in the context of CD40 and IL4 stimulation. To confirm that the apparent selective effect of PU.1 on gene expression was not limited to the single stimulus condition of CD40 and IL4, it would be valuable to address differential gene expression in the context of TLR driven activation/differentiation at a similar time point. This could help substantiate the conclusion that the role of PU.1 is general and not stimulus specific.

RESPONSE: We have performed RNAseq on B cells stimulated in LPS+IL-4. These data shown in Fig. S6 and S7 show a similar hyper-differentiation phenotype in PU.1 KO and PU.1/SpiB DKO to that observed in CD40L+IL-4. Also similar is the relatively low cellularity of DKO cultures in LPS+IL-4, CD40L+IL-4 or anti-IgM+IL-4. Analysis of the key target genes identified in this study showed that the decreased Cd40, Tlr4, Tlr9 and Tnfrsf13c (BAFFR) and multiple components of the BCR pathway in the absence of PU.1/SpiB (Fig. S7) was not stimulus dependent. These findings and the in vivo lack of GCs after protein immunization suggest a common role for PU.1/SpiB in B cell differentiation in response to multiple signals.

12) In Figure 7 there is a link drawn between differential gene expression and impaired BCR signaling with loss of expression of many components in the PU.1 and PU.1/SPIB double deficient setting. In Figure 7D the SPIB deficient context is described as showing a “modest” reduction in BCR calcium flux with a “severe reduction in the absence of

PU.1 or PU.1 and SPIB” (page 12 line 259-260), and yet this interpretation is not immediately obvious from the data where the major difference appears to be between wild type and any of the other genotypes. One might equally interpret the data shown as supporting a major difference between any of the knockout genotypes and wild type and a modest difference between the SPIB deficient and PU.1 or PU.1/SPIB double deficient cells. In order to relate the BCR signaling defect to the differential gene expression it would therefore be necessary to provide an explanation for the defect in the SPIB deficient context. Furthermore, it would be important to consider the relative phenotypes of the cells responding to BCR ligation, and any impact of accelerated differentiation at the 48h time point of CD40 and IL4 stimulus used (as the same issue could apply as noted above).

RESPONSE: We acknowledge the reviewer’s point here. While measuring Calcium flux after BCR crosslinking shows an impairment in the absence of PU.1 or PU.1/SpiB, there is also reduced signaling without SpiB. Our data clearly show direct PU.1 binding in the promoter of many BCR signaling genes and that the combination of PU.1/SpiB regulates multiple BCR signaling pathway components, while the mechanism for the reduced Calcium flux without SpiB is less obvious. Potentially PU.1 and SpiB regulate the distinct BCR pathway genes both individually and redundantly and thus multiple distinct defects could lead to the same reduced signaling phenotype. Importantly our immunization data (Fig. 3) shows that GC formation and function occurs relatively normally in the absence of SpiB alone, while there is a partial defect in PU.1 KO and a complete loss of GCs in PU.1/SpiB DKO. Thus, it is only when both factors are absent that the strongest and most immunologically relevant phenotype is observed.

See Fig. S3 and the response above for our analysis of the differentiation status of the activated B cells at 48hrs.

REVIEWERS' COMMENTS:

Reviewer #1 (Remarks to the Author):

The authors have addressed all my technical comments.

Reviewer #2 (Remarks to the Author):

The authors have responded satisfactorily to all the issues raised in my initial review.

Reviewer #3 (Remarks to the Author):

The additional data shown and the arguments made in rebuttal have substantially addressed the points raised, I think this is an interesting and valuable manuscript.